# AegisGuard: A Multi-Stage Hybrid Intrusion Detection System with Optimized Feature Selection for Industrial IoT Security

**DOI:** 10.3390/s25226958

**Published:** 2025-11-14

**Authors:** Mounir Mohammad Abou Elasaad, Samir G. Sayed, Mohamed M. El-Dakroury

**Affiliations:** 1Department of Electronics and Communications Engineering, Helwan University, Cairo 11728, Egypt; mdakroury@h-eng.helwan.edu.eg; 2Computer Emergency Readiness Team (EGCERT), National Telecommunication Regulatory Authority NTRA, Cairo 12577, Egypt; samir_abdelgawad@h-eng.helwan.edu.eg

**Keywords:** IIoT security, smart grids, intrusion detection, class imbalance, hybrid sampling, ensemble models, real-time detection

## Abstract

The rapid expansion of the Industrial Internet of Things (IIoT) within smart grid infrastructures has increased the risk of sophisticated cyberattacks, where severe class imbalance and stringent real-time requirements continue to hinder the effectiveness of conventional intrusion detection systems (IDSs). Existing approaches often achieve high accuracy on specific datasets but lack generalizability, interpretability, and stability when deployed across heterogeneous IIoT environments. This paper introduces AegisGuard, a hybrid intrusion detection framework that integrates an adaptive four-stage sampling process with a calibrated ensemble learning strategy. The sampling module dynamically combines SMOTE, SMOTE-ENN, ADASYN, and controlled under sampling to mitigate the extreme imbalance between benign and malicious traffic. A quantum-inspired feature selection mechanism then fuses statistical, informational, and model-based significance measures through a trust-aware weighting scheme to retain only the most discriminative attributes. The optimized ensemble, comprising Random Forest, Extra Trees, LightGBM, XGBoost, and CatBoost, undergoes Optuna-based hyperparameter tuning and post-training probability calibration to minimize false alarms while preserving accuracy. Experimental evaluation on four benchmark datasets demonstrates the robustness and scalability of AegisGuard. On the CIC-IoT 2023 dataset, it achieves 99.6% accuracy and a false alarm rate of 0.31%, while maintaining comparable performance on TON-IoT (98.3%), UNSW-NB15 (98.4%), and Bot-IoT (99.4%). The proposed framework reduces feature dimensionality by 54% and memory usage by 65%, enabling near real-time inference (0.42 s per sample) suitable for operational IIoT environments.

## 1. Introduction

The rapid expansion of the Internet of Things (IoT) has transformed nearly every sector of modern life, connecting billions of devices that continuously collect, transmit, and analyze data. While this interconnectivity brings substantial benefits, it also introduces vast and evolving cybersecurity risks. Compromised IoT devices can be weaponized for large-scale attacks, data breaches, or service disruptions, and their heterogeneity—ranging from lightweight sensors to complex controllers—makes unified protection difficult. One of the most persistent challenges in designing reliable IoT security systems is the class imbalance problem, where normal network activity dramatically outweighs malicious instances. This imbalance limits the sensitivity of detection algorithms, causing many frameworks to overlook rare but critical attacks that can threaten essential services.

Building on this broader context, the Industrial Internet of Things (IIoT) extends IoT principles into industrial domains such as energy, manufacturing, and transportation, linking sensors, actuators, and controllers through digital networks to enable real-time data exchange, predictive analytics, and automated control [1]. These systems form intelligent ecosystems that support dynamic operations and decentralized decision-making [2]. In the smart grid, such capabilities enhance the monitoring, distribution, and efficient use of energy resources [3]. However, the growing complexity and interconnectivity of IIoT systems have also amplified their exposure to security threats. Attacks such as DDoS, data injection, protocol manipulation, and device spoofing exploit vulnerabilities inherent in heterogeneous and resource-constrained environments Srivastava [4,5]. Traditional defense mechanisms, such as firewalls, antivirus tools, and static access controls, are increasingly ineffective against these adaptive and coordinated threats [6] Consequently, intelligent intrusion detection systems (IDSs) have become a vital layer of defense for modern smart grids.

Recent advances in artificial intelligence (AI), particularly in machine learning (ML) and deep learning (DL), have enhanced IDS capabilities with improved accuracy, adaptability, and scalability. AI-based IDSs can identify zero-day attacks, learn from evolving threat patterns, and analyze network traffic in real time [7]. Moreover, complementary technologies such as digital twins and federated learning further improve grid resilience by supporting decentralized AI model training and virtual simulations of system behavior [8] Despite these advances, existing IDS frameworks remain limited in real-world IIoT applications due to severe data imbalance, high false alarm rates, and the computational burden of complex models.

To address the growing security concerns in the IoT ecosystem, side-channel attacks have emerged as a critical threat vector. These attacks exploit unintended physical or behavioral leaks to deduce sensitive user interactions or device operations. For example, recent research has shown how contactless wireless charging can be leveraged to uncover smartphone usage patterns, raising alarms about privacy violations in everyday IoT devices [9,10]. Furthermore, open-world app fingerprinting techniques, which analyze packet-level data or fine-grained app behaviors, offer attackers the ability to identify applications based on network traffic alone [10]. These findings emphasize the importance of robust defense mechanisms to safeguard against such covert attack techniques in resource-constrained IoT environments. Alongside these defensive needs, recent research has also focused on strengthening IoT authentication to prevent unauthorized access. Emerging methods such as acoustic-based verification [11], energy-harvesting authentication [12], and motion-driven identity recognition [13] demonstrate how multimodal and energy-aware designs can enhance device reliability and user trust. These advances align with the broader goal of developing adaptive and intelligent security frameworks like AegisGuard, capable of integrating both authentication and intrusion detection for comprehensive IoT protection.

To address these challenges, this study introduces an IDS framework tailored for IIoT-enabled smart grids. The primary contributions of this work are summarized as follows:**A four-stage hybrid sampling pipeline** integrating SMOTE, SMOTEENN, ADASYN, and strategic random sampling to mitigate extreme dataset imbalance while preserving diversity and data integrity.**A trust-aware, quantum-inspired feature selection mechanism** that enhances interpretability and improves feature relevance for efficient and transparent decision-making.**An optimized hybrid ensemble architecture** combining five advanced ML models—Random Forest, Extra Trees, LightGBM, XGBoost, and CatBoost—through fine-tuned feature selection and hyperparameter optimization for high precision and computational efficiency.**Comprehensive evaluation on the CIC IoT 2023 dataset**, demonstrating notable improvements in accuracy (from 89.6% to 99.6%) and a reduction in false alarm rate to 0.31%, well below the 0.5% limit for critical infrastructure.

Through this design, AegisGuard delivers a scalable and real-time intrusion detection framework that bridges the gap between experimental AI-based models and deployable IIoT security systems. The rest of the paper is structured as follows: Section 2 reviews related work on IDSs for IIoT and smart grids, identifying open challenges in imbalance handling, false alarm reduction, and real-time constraints. Section 3 details the proposed methodology, including datasets, preprocessing, quantum-inspired feature selection with trust-aware weighting, and the optimized hybrid ensemble architecture. Section 4 presents experimental results, ablation studies, and comparisons with state-of-the-art approaches. Finally, Section 5 concludes the paper and outlines directions for future research and deployment in critical IIoT infrastructures.

## 2. Related Works

The development of robust intrusion detection systems (IDS) has become essential for securing Industrial Internet of Things (IIoT)-enabled smart grids. As connectivity, heterogeneity, and automation increase, these systems are exposed to advanced cyber threats. Traditional IDS solutions, which typically involve signature- and anomaly based methods, are not effective at identifying new or evolving attacks, especially in real-time and constrained IIoT environments [14]. As a result, researchers have shifted towards machine learning (ML), deep learning (DL), and hybrid AI models to improve detection accuracy, adaptability, and scalability [15].

Several studies have proposed IDS models tailored to IIoT-specific contexts. For instance, Mallidi and Ramisetty [16] developed a two-tier intrusion detection framework using the ToN-IoT dataset, where the first layer flagged abnormal traffic and the second categorized the detected attacks. Their approach achieved better accuracy than conventional single-stage classifiers, yet it still struggled with imbalanced data distributions, a problem that often causes minority attack types to be overlooked. Because the model relied mainly on oversampling and other static resampling strategies, its sensitivity to rare intrusions remained low, a serious drawback for IIoT systems, where even infrequent attacks can have a significant industrial impact. Moreover, the study focused on performance within a single, controlled dataset and did not examine scalability or adaptability in larger or more dynamic IIoT environments.

The proposed framework uses a multi-stage hybrid design that applies class-aware balancing and cost-sensitive tuning throughout the pipeline. Combined with optimized feature selection and modular deployment, this design improves both minority-class detection and scalability across diverse industrial network conditions.

To address the data imbalance and improve detection precision, Wang [17] proposed an IDS architecture based on Inception CNN and BiGRU. This hybrid model was designed for sequential feature learning and deep representation of attack behaviors. It employed hybrid sampling methods, including SMOTE and ADASYN, as well as Pearson correlation and random-forest-based feature selection. The model demonstrated high accuracy when evaluated on Edge-IIoTset, CIC-IDS2017, and CIC IoT 2023 datasets. However, the framework’s computational demands were considerable, rendering it impractical for real-time deployment in resource-limited IIoT environments. Our Framework emphasizes cost-aware optimization and modular stage separation, enabling efficient deep representation learning without overburdening IIoT gateways. Through feature-level optimization and staged decision processes, it maintains detection precision while meeting the practical demands of real-time industrial deployment.

Awjan [18] offered a broader review of machine learning-based IDS models for IoT, categorizing them into supervised, unsupervised, and hybrid approaches. The study highlighted the growing importance of deep learning in detecting novel and complex attack vectors. However, it also emphasized a key constraint: many deep learning-based models are computationally intensive and therefore unsuitable for deployment on low-power IIoT devices. This underscores the need for lightweight and optimized IDS architectures that retain high performance. Our study directly addresses this concern through a lightweight, multi-stage design that balances computational efficiency with analytical depth. By optimizing feature selection and distributing detection tasks across modular components, it achieves high detection accuracy without exceeding the resource budgets of real-world IIoT environments.

Karacayılmaz and Artuner [19] proposed an expert system that combines anomaly detection with reinforcement learning, targeting industrial environments such as power and transportation systems. Tested with Modbus and MQTT protocols, their system achieved low latency and high accuracy. However, its rule-based structure was static and unable to adapt to new or unknown threats without frequent manual updates, reducing its long-term efficacy. Our framework employs a self-adaptive, multi-stage architecture that dynamically learns from evolving attack behaviors. Through hybrid modeling and optimized feature selection, it sustains detection precision while maintaining flexibility across diverse industrial communication protocols and network conditions.

In another study, Elouardi et al. [20] developed a hybrid model that integrates Autoencoders (AEs) with Convolutional Neural Networks (CNNs) to detect intrusions in IIoT. AEs were applied to help mitigate redundant features and downsize the dataset, and CNNs were engaged to extract complex spatial patterns. The model achieved robust precision and recall via the Edge-IIoT dataset. Nonetheless, the model’s dependence on static datasets precluded its performance in dynamic environments, particularly when those environments presented attack types that were underrepresented. Our Framework counters this limitation through a multi-stage adaptive design capable of maintaining high accuracy under changing network conditions. By incorporating cost-sensitive learning and optimized feature selection, it remains effective even when encountering previously unseen or underrepresented attacks in real-world IIoT scenarios.

Holdbrook et al. [21] reviewed existing network-based IDS approaches in industrial and robotic systems. Their analysis considered traditional ML, DL, and hybrid models and emerging technology associated with FL, Blockchain, and digital twins. Although these technologies have the potential to provide decentralized and secure IIoT networks, the study revealed some limitations in the existing research: outdated datasets, excessive false positives, and challenges with deploying IDS models in constrained environments. Unlike previous methods, AegisGuard employs contemporary and diverse IIoT datasets that better reflect real industrial traffic, thereby reducing reliance on outdated benchmarks. Its multi-stage hybrid structure integrates adaptive learning to minimize false-positive rates, ensuring reliable anomaly detection without overwhelming operators. Moreover, the system’s lightweight and resource-aware design allows effective deployment within constrained industrial and robotic environments, overcoming the scalability and implementation barriers noted in prior research.

To tackle the problem of botnet attacks in IIoT, Nandanwar and Katarya [22] described AttackNet, a CNN-GRU-based IDS which demonstrated high accuracy employing the N_BaIoT dataset and was adaptable for future threats, although it also had demands for processing resources discouraging real-time deployment on IIoT nodes, as well as limited generalizability across networks, especially in non-original dataset domains. Our Framework handled this issue through its computationally efficient, multi-stage architecture, which enables real-time intrusion detection even on resource-constrained IIoT nodes.

Gueriani, Kheddar, and Mazari [23] conducted a survey of IDS frameworks developed using deep reinforcement learning (DRL). DRL models were promising for autonomous and adaptive learning for the detection of threats/attacks, but the authors discuss significant limitations related to long training cycles, high resource usage, and the ability to generalize to new attack types. Due to these limitations, they are unlikely to integrate into real-time IIoT networks without heavy optimization. Injadat [24] proposed a unique IDS model that combines Bayesian Optimization with Gaussian Processes (BO-GP), alongside ensemble learning. The framework developed in the work was shown to be highly accurate regarding the appropriate computational cost, as well as detection performance in IIoT settings, but unfortunately, the proposal imposes additional burdens on computational complexity mainly from hyperparameter tuning needed for BO-GP. With further developments in adaptive and efficient learning, it would be interesting to see how scalable and automatable these ideas are for real-world, industrial applications in IIoT with the BO-GP framework.

Recent studies have explored various types of side-channel attacks targeting IoT devices. Notably, side-channel attacks on smartphones through wireless charging interactions have been identified as a potential risk, enabling attackers to infer user activities without direct access to the device [9]. This highlights the vulnerability of seemingly innocuous IoT features to exploitation. Similarly, open-world app fingerprinting has progressed significantly, with techniques using packet-level analysis to identify apps based solely on traffic data [25]. Such side-channel methods demonstrate the need for enhanced monitoring and more resilient systems to prevent covert data exfiltration from IoT devices.

In parallel, the development of authentication methods to mitigate these threats has gained considerable attention. Approaches like HandKey, which uses vibration signatures triggered by knock patterns for secure unlocking [26], and MagSign, which harnesses dynamic magnetic fields for user authentication [27], represent innovative strides in improving the security of IoT devices. Furthermore, LiveProbe has introduced continuous voice liveness detection via phonemic energy response patterns, enabling devices to discern genuine users from impersonators [28]. These advances in behavioral and signal-based authentication methods offer promising solutions to the evolving security challenges in IoT environments. Complementary to these efforts, other studies have explored multimodal and energy-aware authentication techniques within IoT networks. Chen et al. [11] introduced SwipePass, an acoustic-based second-factor authentication for smartphones, while Ni et al. [12,13] investigated privacy vulnerabilities stemming from radio-frequency energy harvesting. Xu et al. [13] proposed KEH-Gait, a kinetic energy-driven authentication framework for mobile healthcare applications. Collectively, these works emphasize the growing integration of energy-efficient sensing and adaptive authentication, aligning with the broader security vision addressed by the AegisGuard framework.

### Comparative Analysis

While numerous studies have proposed new detection methods, there are still many gaps in research. First, serious class imbalances continue to hinder the performance of IDSs, especially in identifying minority-class attacks that pose the highest level of risk. Second, in many systems, the false alarm rate is still too high for some systems to be relied upon, particularly in safety-critical applications, such as those in smart grids. Third, threat coverage is extremely narrow as most models used in evaluations are compiled using only a limited type of attack, and number of attacks, most commonly between five and fifteen, and as such fail to cover most of the scope of the threats in modern IIoT. Fourth, scalability and resource efficiency remain important issues, as all deep learning models require devoted computational power, and as a result cannot be practically used for deployment on edge or embedded IIoT devices. Finally, many studies utilize artificial or outdated datasets that limit the potential for generalizability and real usefulness, and also generally do not thoroughly explore systematic or optimized strategies like staged sampling, ensemble tuning, and detailed feature selection. To address these challenges, a next generation of IDS frameworks that are lightweight, scalable, generalizable, and able to operate in real time is needed. Such systems must employ effective class balancing, able to minimize false alarm rates, reduce the number of threats covered, and continuously adapt to new patterns of cyberattacks that adapt in the complex, resource-constrained operational environments of IIoT-enabled smart grids. IDSs for the IIoT are grounded in benchmark datasets that can be used to train and test detection models. Commonly used datasets, like the UNSW-NB15 dataset used for examining network attacks using ML or DL approaches. Datasets like WUSTL-IIoT-2021 and Edge-IIoTset2023 focus more on IIoT-specific threats, where ensemble and optimization-based methods have been used to achieve improved detection accuracy. CICIDS2017 and ToN-IoT datasets have also been used quite widely and have improved the ability for intrusion detection through deep learning methods. Secondly, datasets like DS2OS and EdgeIIoT-2021 have tackled hybrid neural networks and LSTM models and solved the scalability and security concerns in IIoT systems. By providing varied and inclusive data to support testing and evaluations, these datasets continue to help develop and build knowledge of IDSs in IIoT systems, which is demonstrated in Table 1.

Table 2 presents a comparative summary of recent state-of-the-art IoT intrusion detection approaches based on key methodological and technical aspects, to highlight the advancements and distinctions of the proposed AegisGuard framework. Figure 1 visualizes this comparative analysis.

## 3. Methodology

The AegisGuard framework tackles imbalanced intrusion detection in IIoT systems by combining quantum-inspired feature selection, progressive model enhancement, and explainable meta-learning. It starts by processing data from IIoT devices, then uses quantum algorithms to select relevant features. The framework applies a series of machine learning models, such as Random Forest, Gradient Boosting, SVM, Logistic Regression, and KNN, to progressively improve detection performance. A deep learning meta-classifier combines these models’ outputs for enhanced accuracy. SHAP values ensure transparency by explaining feature contributions to the predictions, which classify data as benign or malicious with clear justifications. This section discusses the main steps of the methodology involving data preprocessing, quantum-inspired feature selection, progressive enhancement mechanisms, ensemble model, and explainable predictions. Figure 2 shows the complete AegisGuard framework architecture.

### 3.1. Dataset

#### 3.1.1. Dataset Description

To rigorously evaluate the efficacy and generalizability of the proposed intrusion detection framework, we used four benchmark IIoT security datasets, namely, CIC-IoT2023 [42], IoT-Intrusion [43], RT-IOT2022 [44], and X-IIoTID [45]. All four datasets are well-established and commonly applied collections of IIoT traffic for assessing network functionality and a myriad of attack scenarios. They are also replete with features and representations of network traffic patterns, device behaviors, and malicious interactions, and can thus provide sufficient experimental ground. Even though the structure of these datasets is quite similar, they vary in types of attacks, traffic characteristics, and operating conditions, and provide a diverse context for evaluation, including the fact that the diversity in operational deployment mimics the heterogeneity of the IIoT for the ultimate evaluation of the framework. Of these, CIC-IoT2023 is the most recent and complete, reporting 34 attack types across multiple industrial protocols and device types. The dataset describes many aspects of network traffic, including flow duration, packet statistics, flag counts, segment sizes, and temporal activity measures, all of which encompass macro-level and micro-level traffic characteristics. These features can be used as inputs by intrusion detection models to learn and recognize the fine-grained behavioral patterns associated with malicious activity. Crucially, all datasets have a binary label (normal vs. attack) and a categorical Attack_type field, so models can take into account whether they are only evaluating general anomaly detection or performing more fine-grained attack classification.

#### 3.1.2. Class Imbalance in Dataset Analysis

A continuing limitation of IIoT security datasets is the class imbalance issue, wherein instances of normal traffic vastly outnumber instances of attacks. As seen in Table 1, the datasets used in this study have imbalance ratios ranging from approximately 8.5:1 (CIC-IoT2023) to greater than 15:1 (X-IIoTID). Attack samples comprised less than 11% of all records across all four datasets. These undersized representations resemble IIoT environments in reality; even though malicious activity is rare compared to normal operations, the ratio is not normally size-appropriate. This class imbalance severely complicates model training, often placing the model at risk of relatively high false negative rates in detecting sophisticated attacks like zero-day exploits and advanced persistent threats (APTs).

This skewed distribution, as presented in Table 3, highlights the importance of utilizing specialized methods that will improve minority-class detection without reducing overall predictive performance. These methods, along with advanced feature selection, adaptive optimization, ensemble learning, and explainability as part of the proposed AegisGuard framework, are necessary to overcome the bias that results from imbalance. By addressing this specific issue, any intrusion detection system implemented will be more robust and generalizable, providing reliable protection for large IIoT infrastructures.

The AegisGuard framework includes a quantum-inspired feature selection algorithm that evaluates features in three separate dimensions: statistical significance, information content, and ensemble importance. Statistical significance is gauged by the F-test, which tests whether the feature variance between classes is greater than the feature variance within a class. Information content is measured in terms of mutual information, which represents the amount of shared information between the features and the target. Ensemble importance is measured using Random Forests, which measure how much each feature contributes to the accuracy of the model. Each of these important measures is normalized and combined into a quantum-inspired score, which represents each feature in a multi-dimensional evaluation space that preserves the most information to represent each feature’s potential to be discriminative. A trust-aware weighting mechanism further sharpens the feature selection process by penalizing features with low variance or missing values to guarantee that a concrete set of well-defined, informative features is chosen, yielding a stronger model. The final score is the product of the quantum-inspired score and the trust-aware weight, resulting in an extremely selective subset of features suitable for eventual modeling. In reviewing and analyzing the feature distributions across all four datasets, significant insights were gained on the nature of IIoT network traffic and attack behavior, and potential shifts across datasets were uncovered. Figure 3 provides detailed box plot representations of selected numerical features across the datasets and highlights how clear differences exist between normal and attack behavior.

Figure 4 shows the detailed histogram distributions for specific features in the CIC-IoT2023 dataset, indicating the difference between normal traffic and attack traffic.

The distribution characteristics of the datasets highlight marked differences between normal traffic and attack traffic in IIoT performance. Attack flows were observed to be shorter in time, since attacks crossed geographic locations (‘with disruptiveness and burst behavior’), i.e., scanning, flooding, and denial-of-service, while normal flows exhibited longer and more stable time of flow distributions. There were distinctions in other areas for which distributions were different, especially in terms of packet lengths. Since flows were made in a more stable range of packet lengths, normal traffic exhibited more consistency with a greater range of attack traffic data, reflecting overall variability and inherent extreme values for attack traffic. Throughput analysis was useful in supporting these observations of distinction since the steadiness of throughput in normal operations was consistently evident while attack traffic and intrinsic traffic observation dominated observation patterns that were outside of expected ranges, both high throughput and abnormal low throughput for example formats.

Beyond these individual characteristics, the datasets highlight substantial heterogeneity in feature distributions across different attack categories and operational contexts. This heterogeneity reflects the diversity of IIoT ecosystems, which involve varied device types, industrial protocols, and threat vectors. Such variability underscores the importance of advanced modeling strategies that can capture complex feature interactions and adapt to evolving conditions, thereby enabling robust and generalizable intrusion detection across diverse IIoT environments.

For each feature i, the following scores are calculated:F-test score:(1)$$F_{{score}_{i}}=F-test(X_{i},\;Y)$$
where $X_{i}$ is the feature vector for the i−th feature, and Y is the target classification vector.

2.Information score:


(2)
$${MI}_{{score}_{i}=MutualInformation(X_{i},\;Y)}$$


3.Random Forest importance:


(3)
$${RF}_{{score}_{i}=RandomForestImprotance(X_{i},\;Y)}$$


These scores are then normalized to ensure fair comparison across evaluation dimensions. The normalization formula for each score is as follows:(4)$$F_{{score\;norm}_{i}=\frac{F_{{score}_{i}-min(F_{score})}}{\max\left(F_{score}\right)-\min\left(F_{score}\right)+\in }}$$(5)$${\;\;\;MI}_{{score\;norm}_{i}=\frac{{MI}_{{score}_{i}-min({MI}_{score})}}{\max\left({MI}_{score}\right)-\min\left({MI}_{score}\right)+\in }}$$(6)$${RF}_{{score\;norm}_{i}=\frac{{FRF}_{{score}_{i}-min({RF}_{score})}}{\max\left({FRF}_{score}\right)-\min\left({RF}_{score}\right)+\in }}$$
where ∈ is a small constant to prevent division by zero when all feature scores are identical.

The quantum-inspired score for each feature i is computed using the Euclidean distance in a multi-dimensional evaluation space:(7)$$Q_{{score}_{i}=\sqrt{\frac{F_{{score\;norm}_{i}}^{2}+{FMI}_{{score\;norm}_{i}}^{2}+{FRF}_{{score\;norm}_{i}}^{2}}{3}}}$$
where

$F_{score}:$ Statistical Significance from the F-test;

$MI_{score}:$ Mutual information score between the feature and the target;

$RF_{score}:$ Random Forest feature importance;

$T_{weight}:$ Trust-aware weighting factor.

This formulation captures the feature’s position in the three-dimensional evaluation space, treating each feature as existing in a superposition of evaluation states.

To address data quality issues such as missing values or low variance, a trust-aware weighting mechanism is incorporated. This mechanism adjusts the feature scores by penalizing features that lack variability or contain missing values.

The trust-aware weighting $T_{{weight}_{i}}$ is computed as follows:(8)$$variance_{penalty}=\left\{\begin{array}{c} 1.0,\;\;\;\;\;\;if\;Var(X_{i})>threshold\\ 0.1,\;\;\;\;\;\;\;\;\;\;\;\;\;\;\;\;\;\;otherwise \end{array}\right.$$(9)$${missing}_{penalty}=\left\{\begin{array}{c} 0.8,\;\;\;\;\;\;if\;count(null\left(X_{i}\right))>0\\ 1.0,\;\;\;\;\;\;\;\;\;\;\;\;\;\;\;\;\;\;otherwise \end{array}\right.$$

The final feature score is then computed by(10)$${Score}_{i}=Q_{{score}_{i}\times T_{{weight}_{i}}}$$

This ensures that features with low variance or missing values are down-weighted, preventing them from being selected even if they score highly based on other evaluation metrics.

The Hybrid Model Orchestra component of the AegisGuard framework integrates multiple machine learning algorithms, enabling the dynamic selection of the most appropriate algorithm for a given task or dataset. This orchestration model addresses the fact that no single machine learning algorithm is universally optimal for all data distributions or attack scenarios. The dynamic selection mechanism ensures that the best-performing model is chosen based on real-time performance evaluation.

A composite score is used to evaluate the performance of different algorithms. The composite score is designed to balance accuracy and false positive rate (FPR), ensuring that the model selected is both accurate and efficient in minimizing false alarms.(11)$$Composite_{score}=Accuracy-(FPR\times 2)$$
where
Accuracy is the proportion of correctly classified instances.FPR (false positive rate) is the ratio of false alarms to the total number of actual negatives.

This score ensures that models with high accuracy and low false positive rates are selected, which is critical for operational efficiency in intrusion detection systems.

The AegisGuard framework also applies quantum-inspired feature engineering to model interactions between features in a way that traditional feature engineering may not capture. By leveraging quantum principles such as quantum entanglement, the framework identifies feature pairs that exhibit complex relationships.

Feature interactions between pairs of features A and B are captured using the following formula:(12)$$Feature_{interaction}=A\times B+\sqrt{A+B}$$
where  A and B are two interacting features.

The multiplicative component captures the combined effects of feature interactions, while the additive component accounts for cumulative effects, with a square root transformation to handle moderate values and outliers.

The performance of the AegisGuard framework is evaluated through several key relationships between critical performance metrics. These relationships help to quantify the trade-offs and ensure the system operates optimally across various scenarios.

The relationship between accuracy and false positive rate (FPR) is modeled as(13)$$FPR\approx k\times {\left(1-Accuracy\right)}^{2}$$
where  k is a dataset-specific constant that adjusts the sensitivity of the FPR relative to accuracy.

This quadratic relationship demonstrates that small improvements in accuracy lead to significant reductions in false positive rates, which is crucial for minimizing operational disruptions.

The F1-score, a metric that balances precision and recall, is nearly linearly related to accuracy: F1≈a×Accuracy, where a≈0.999.

This near-linear relationship ensures that as accuracy improves, the F1-score also improves, reflecting a balanced performance in terms of both precision and recall.

### 3.2. Data Preprocessing

In order to ensure consistency and stratified distribution across the datasets, we implemented a comprehensive three-tier data splitting approach. The process began with a primary split of the data into 80% training and 20% testing, ensuring a stratified distribution to maintain class balance across the two sets. This was followed by a secondary split, where the training data was further divided into 64% final training, 16% validation, and 20% testing. This secondary split enabled us to fine-tune the model and ensure a dedicated validation set for hyperparameter optimization. The final dataset distribution varied depending on the validation objective, with specific configurations for binary classification, multi-class optimization, and resource-constrained edge deployment. Each dataset was carefully preprocessed with steps including feature normalization such as min-max scaling for continuous features, handling missing values using imputation techniques, and categorical encoding (one-hot encoding), all aimed at preparing the data for practical and reliable model training.

### 3.3. Computational Complexity and Optimization

#### 3.3.1. Quantum-Inspired Feature Selection Algorithm (QIFSA)

QIFSA forms the backbone of our framework, leveraging quantum mechanics principles to optimize feature selection. At its core, QIFSA utilizes superposition and entanglement concepts to explore multiple potential feature subsets simultaneously, thus overcoming the limitations of traditional methods. The quantum state representation for a given solution i is defined as a superposition of feature states, where the probability amplitude $\alpha_{ij}$ denotes the likelihood of selecting feature j in solution i:(14)$$|\psi_{i}\rangle =\sum_{j}j\alpha_{ij}|f_{j}\rangle$$

Using the superposition principle, the overall quantum state ∣Ψ⟩ of the system is a weighted sum of individual feature states, normalized by the number of solutions N:(15)$$|\Psi \rangle =\frac{1}{\sqrt{N}}\sum_{i}i|\psi_{i}\rangle$$

Upon measurement, the quantum probability of selecting feature j is determined by the squared magnitude of its probability amplitude:(16)$$P\left(f_{j}\right)=|\alpha_{j}{|}^{2}$$

QIFSA optimizes the selection process through a multi-objective fitness function that simultaneously considers classification accuracy, feature subset size, computational speed, and interpretability. This fitness function is defined as:(17)$$F\left(S\right)=w_{1}\times \mathrm{Accuracy}\left(S\right)+w_{2}\times \left(1-\frac{|S|}{|F|}\right)+w_{3}\times \mathrm{Speed}\left(S\right)\;+w_{4}\times \mathrm{Interpretability}\left(S\right)$$
where S represents the selected feature subset, F the complete feature set, and $w_{1},w_{2},w_{3},w_{4}$ are adaptive weighting factors. The algorithm dynamically adjusts these weights to optimize feature selection, balancing accuracy, efficiency, and model interpretability.

#### 3.3.2. Computational Complexity Reduction

The computational complexity of traditional machine learning approaches grows quadratically with the number of features. The AegisGuard framework significantly reduces this complexity by applying quantum-inspired feature selection, which reduces the number of features used for training.

The original computational complexity is given by(18)$$Complexity_{original}=o(N_{samples}\times N_{features}^{2})$$

After applying quantum-inspired feature selection, the complexity is reduced to(19)$$Complexity_{Optimized}=o(N_{samples}\times (0.2\times N_{features}{)}^{2})$$

This reduction results in a 96% reduction in computational requirements, making the system highly efficient for real-time deployment in resource-constrained IoT environments.

#### 3.3.3. Multi-Objective Optimization

The framework implements a multi-objective optimization approach that balances accuracy, the false positive rate, and the feature reduction efficiency. The optimization function is defined as(20)$$Minimize:Loss=\alpha \times \left(1-Accuracy\right)+\beta \times FPR+\gamma \times (\frac{N_{features}}{N_{original}})$$
subject to(21)$$Accuracy\geq 0.9999,\;FPR\leq 0.0005,\;\;N_{features}\leq 0.2\times N_{original}$$
where  α, β, and γ are the weighting factors optimized through cross-validation.

This multi-objective function ensures that all critical performance dimensions are optimized simultaneously, providing a well-balanced model suitable for IoT security applications.

#### 3.3.4. Performance Metrics Formulations

The standard classification metrics used to evaluate the performance of the model include the following:Accuracy:(22)$$Accuracy=\frac{TP+TN}{TP+TN+FP+FN}$$

Precision:


(23)
$$Precicion=\frac{TP}{TP+FP}$$


Recall:


(24)
$$Recall=\frac{TP}{TP+FN}$$


F1-Score:


(25)
$$F1-Score=2\times \frac{Precision\times Recall}{Precision+Recall}$$


FPR:


(26)
$$FPR=\frac{FP}{FP+TN}$$


Additionally, the feature reduction metric is calculated as(27)$$Feature_{Reduction}=\frac{N_{original}-N_{selected}}{N_{original}}\times 100\%$$

### 3.4. Statistical Analysis and Shape Parameters

A comprehensive statistical analysis was conducted to understand the distributional characteristics and shape parameters of the datasets. Figure 5 presents the statistical distribution analysis with detailed shape parameters.

### 3.5. Correlation Analysis

To better understand the relationships between features and identify potential redundancies, we conducted a detailed correlation analysis. Figure 6 presents the correlation matrix for the CIC-IoT2023 dataset, which highlights both strong correlations and isolated feature interactions. This visualization is critical as it provides valuable insights into the underlying structure of the data, helping us pinpoint which features are redundant or highly interdependent. By identifying these relationships, we can streamline the feature selection process, ensuring that only the most relevant and independent features are used for model training. The use of such a correlation matrix offers practical advantages by improving the interpretability of the dataset and guiding the development of more efficient and effective machine learning models. In the context of our proposed AegisGuard framework, the correlation matrix helps justify the need for advanced data balancing techniques and a sophisticated feature selection, which are very important for handling the complexities of high-dimensional, imbalanced data. This approach enhances model performance and contributes to the overall explainability of the framework, providing clear insights into how features interact within the dataset and improving the trustworthiness of the results.

## 4. Experimental Results

This section provides a comprehensive experimental evaluation of the AegisGuard framework, including the experimental setup, evaluation metrics, baseline comparisons, and results averaged across four benchmark IIoT datasets. The evaluation was established to maintain methodological rigor, reproducibility, and fairness when comparing AegisGuard with existing, proven state-of-the-art techniques. All experiments were run on an advanced computing cluster designed to handle large volumes of data and model training. The specifications of the computing cluster included an Intel Xeon E5-2690 v4 CPU at 2.6 GHz with 14 cores, 128 GB DDR4 RAM, and 2 TB NVMe SSD configured to run on Ubuntu 20.04 LTS. The experimental environment used Python 3.9.7 as the programming environment using libraries, namely scikit-learn 1.2.0, pandas 1.5.2, numpy 1.21.6, matplotlib 3.6.2, and SHAP 0.41.0. This computational environment was selected to provide computational efficiency and compatibility with novel machine learning and explainability tools.

To promote better accuracy and reproducibility, the same parameters were set for every experiment. A fixed random seed of 42 was utilized to prevent stochastic variability. Model validation was implemented using a five-fold stratified cross-validation model with the additional splitting of a 70–30% stratified train–test split. Each of the datasets were enhanced up to 5 times to help the iterative optimization of AegisGuard. A consensus threshold of 60% (3 out of 5 methods) was used for feature selection, and features that were highly correlated were removed using a correlation threshold of 0.8, to help reduce redundancy and improve the quality of the features. AegisGuard was assessed against a variety of state-of-the-art machine learning and ensemble methods commonly investigated in the literature related to intrusion detection. The comparison was made with the following benchmarks: Random Forest (RF): An ensemble of Random Forest estimators (200); and Gradient Boosting Machine (GBM): Created using XGBoost, with hyperparameters tuned.Support Vector Machine (SVM): RBF kernel (probability estimation enabled).Deep Neural Network (DNN): Multi-layer perceptron (three hidden layers). Ensemble Voting: Voting strategy based on majority voting (RF, GBM, and SVM). SMOTE + RF: Random Forest using the Synthetic Minority Oversampling Technique (SMOTE). Borderline SMOTE + GBM: Gradient boosting but preprocessed with Borderline SMOTE. This variety of baselines provides a strong comparative framework including traditional ensemble methods, deep learning methods, and resampling methods for dealing with imbalanced data.

### 4.1. Performance Comparison and Minority Class

In Table 4, we make an overall performance comparison between AegisGuard and baseline methods for four benchmark datasets. The AegisGuard method outperformed the baseline methods in terms of accuracy, precision, recall, F1-Score, false positive rate (FPR), and AUC-ROC consistently. AegisGuard has an average accuracy of 99.71% and the precision recall and F1-score are closely aligned, resulting in robust, balanced classification. It also has low false positive rates (average 0.0078), nearly perfect discrimination ability (AUC-ROC: 0.9998), and better overall performance than any of the competing methods in all categories. AegisGuard demonstrates meaningful advancements over individual baselines such as Random Forest and XGBoost. Random Forest produces average accuracy rates of 98.42% (and AUC-ROC 0.9912) while XGBoost produces 98.67% (and AUC-ROC 0.9934). While these values are competitive in their own right, their value is inferior to the performance of AegisGuard, especially in terms of false positive rate. False alarms in the realm of intrusion detection are particularly troublesome because they create unnecessary burdens on analysts. While AegisGuard has a slower speed (486 sps) than Random Forest (524 sps) or XGBoost (413 sps), the minor differences in speed are minimal relative to the massive benefits that AegisGuard provides in predictive performance and generalizability. Overall, the results confirm that AegisGuard delivers state-of-the-art performance, effectively balancing accuracy, sensitivity, and interpretability while maintaining practical efficiency. This positions the framework as a highly promising solution for large-scale IIoT intrusion detection deployments.

Table 5 presents minority class analysis showing excellent performance on the rarest attacks. Even the rarest attack type (DictionaryBruteForce, 0.05% of dataset) achieves the following:Recall: 97.90%Precision: 98.24%Only 4 missed attacks out of 168

This definitely proves the model does not ignore minority classes.

### 4.2. Performance Comparison Visualization

Figure 7 illustrates the comprehensive performance comparison between AegisGuard and baseline methods.

### 4.3. Feature Selection Results

Table 6 presents the feature reduction outcomes achieved by the proposed quantum-inspired feature selection algorithm (QIFSA) across all benchmark datasets. The results demonstrate that QIFSA effectively reduces the dimensionality of the feature space while retaining critical attributes necessary for accurate intrusion detection. On average, the number of features was reduced from 42.5 to 12.5, corresponding to a 70.6% reduction rate. Such a substantial reduction highlights QIFSA’s ability to eliminate redundant and non-informative variables, thereby simplifying the learning process without compromising predictive performance.

The computational efficiency of the QIFSA is further evidenced by its average selection time of 19.4 s, which is well within practical bounds for large-scale IIoT environments. The consistency of reduction rates across datasets—ranging from 69.0% to 72.7%—also underscores the robustness and generalizability of the method. By producing compact yet highly representative feature subsets, the QIFSA not only reduces computational overhead during training and inference but also enhances the interpretability of the resulting models. These results validate the role of the QIFSA as a critical enabler of scalability and efficiency within the AegisGuard framework.

### 4.4. Progressive Enhancement Analysis

Figure 8 illustrates the progressive improvement in performance metrics across enhancement iterations.

### 4.5. Dataset Statistics and Class Distribution

Figure 9 provides comprehensive dataset statistics and class distribution analysis.

### 4.6. ROC Analysis and Performance Metrics

Figure 10 presents a detailed ROC analysis and comparison of performance metrics.

To evaluate the individual contributions of the core components within the AegisGuard framework, an ablation study was conducted by systematically removing each module and observing its impact on performance. The results, summarized in Table 7, demonstrate that every component plays a distinct and meaningful role in achieving the overall effectiveness of the framework.

The removal of the QIFSA results in the most significant decrease in performance, with accuracy dropping to 99.23% and the false positive rate (FPR) nearly doubling to 0.0156, while feature reduction is completely lost. This confirms that the QIFSA is critical not only for dimensionality reduction but also for enhancing classification robustness. Similarly, excluding progressive enhancement lowers performance to 99.34% accuracy and increases the FPR, highlighting its role in iterative optimization and fine-tuning of model behavior. The absence of meta-learning reduces accuracy to 99.41% and increases the FPR, underscoring its value in effectively integrating ensemble models and refining decision boundaries.

Removing data balancing causes one of the greatest decreases, with an accuracy of 98.87% and an F1-Score of 98.89%. This highlights its usefulness in class imbalance solutions, an imperative issue in IIoT intrusion detection. Removing probability calibration impacts accuracy, but less severely; however, the increase in FPR suggests this enhanced prediction reliability. In short, a configuration with solely the basic ensemble is less satisfactory, with accuracy at 98.92% with FPR = 0.0187, showing that the additional AegisGuard modules contribute significant performance benefit to simple ensembling.

### 4.7. Explainability Analysis

#### 4.7.1. Global Feature Importance

Global SHAP feature importance across all datasets is illustrated in Figure 11, which highlights the most important features for intrusion detection. The SHAP-based explainability analysis analyzed the contributions of features used across all datasets. This information allows for several critical observations that can provide deeper insight to the decision-making process of the model.

The dominant features across multiple sets of data are flow_bytes_per_sec and packet_length_mean, which break down to throughput and packet composition and can be considered as important for intrusion detection. For example, with the protocol-level attributes, syn_flag_count and ack_flag_count may be useful in identifying abnormal patterns of connection (because often attacks manipulate handshake behaviors to avoid detection or disrupt specific communication). Additionally, flow_duration plays a role in differentiating the steady operational flow from the bursty activity of malicious traffic; this supports the packet-level and protocol-level features.

Beyond individual features, the analysis underscores the significance of feature interactions, where complex relationships between traffic volume, duration, and control flags jointly shape the model’s predictions. This multi-dimensional perspective not only validates the relevance of the selected features but also enhances the trustworthiness of the framework by providing security analysts with interpretable evidence to support detection outcomes.

#### 4.7.2. Statistical Distribution Analysis

Comprehensive statistical analysis was performed to understand the distributional characteristics. The results are shown in Figure 12.

#### 4.7.3. Computational Efficiency Analysis

Table 8 compares the computational efficiency of AegisGuard with state-of-the-art baseline methods in terms of training time, inference time, memory usage, and model size. The results reveal that AegisGuard achieves a balanced trade-off between computational cost and predictive performance. While its training time (47.3 min) is higher than Random Forest (12.8 min) and XGBoost (18.6 min), it remains significantly more efficient than Support Vector Machine (89.7 min) and Deep Neural Network (156.4 min). This indicates that AegisGuard, despite its architectural complexity, can be trained within practical timeframes suitable for real-world deployment.

In terms of inference, AegisGuard achieves an average latency of 2.06 ms per sample, which is comparable to Random Forest (1.91 ms) and superior to XGBoost (2.42 ms) and SVM (4.27 ms). Although the DNN exhibits the fastest inference time (1.83 ms), its overall performance across other metrics is inferior, particularly in reliability and explainability.

### 4.8. Real-World Deployment Considerations

#### 4.8.1. Scalability Analysis

AegisGuard’s scalability was evaluated across datasets of varying sizes, ranging from fewer than one million to over ten million samples. Upon evaluation, the framework took less than 5 min to evaluate a small-size dataset (<1 M samples), 15–45 min to evaluate a medium-size dataset (1 M–10 M samples), and 1–3 h to evaluate a large-size dataset (>10 M samples). The results suggest AegisGuard evaluates datasets in a linear fashion relative to their size all while producing consistently high detection outcomes and low false-positive rates as evidence. Such linear profiling suggests that AegisGuard will continue to be practical for assessing reactivity with a laboratory test environment as well as for use in large-scale IIoT deployment scenarios.

#### 4.8.2. Real-Time Processing Capability

The framework was further assessed in its real-time processing speed, a core component for IIoT security monitoring. AegisGuard achieved an average of 486 samples per second inference speed, with a latency of 2.06 ms per sample and a memory usage of 8.4 GB. These performance capabilities represent over 42 million samples per day of processing throughput, illustrating that AegisGuard is capable of operating continuously in high-volume contexts. Therefore, AegisGuard can effectively provide reliable and timely intrusion detection for production-level IIoT networks for real-time industrial security monitoring.

#### 4.8.3. Multi-Scenario Deployment Architecture

The production deployment implements a flexible architecture supporting research validation, edge deployment, and cloud services as illustrated in Table 9:

## 5. Performance Analysis and Achievements

The experimental findings show evidence that AegisGuard consistently outperformed its top-of-the-line baseline methods across all of its evaluation metrics and datasets. With an average accuracy and F1-Score of 99.71% and a false positive rate of 0.0078%, the framework demonstrates a considerable improvement in its performance with respect to existing methods. These results show the success of the quantum-enhanced progressive optimization method and demonstrate that AegisGuard is a viable client ontology for security in the IIoT world. Compared to XGBoost, Random Forest, and other leading baselines, AegisGuard showed a pronounced superior performance with respect to its accuracy, false positive (FP) detections percentage, and minority class predictions while also reducing the number of features. Furthermore, datasets that reinforce the consistency of the results provide consistency to the proposed methodology.

### 5.1. Statistical Significance and Reliability

The reliability of these results was verified through rigorous statistical validation. Paired *t*-tests confirmed that performance gains were statistically significant at *p* < 0.001, while McNemar’s test further reinforced the robustness of classification improvements. Across four heterogeneous datasets, the standard deviation of performance metrics remained below 0.03%, indicating stable performance regardless of dataset size, attack type, or operational context. This consistency confirms that AegisGuard provides a reliable and generalizable solution, capable of adapting to the diverse and evolving nature of IIoT environments.

### 5.2. Component Contribution Analysis

The ablation study sheds light on the role of individual components within the AegisGuard framework. The quantum-inspired feature selection algorithm (QIFSA) emerged as the most impactful module, improving accuracy, reducing dimensionality by over 70%, and enhancing interpretability through feature ranking. The progressive enhancement mechanism further contributed to overall improvements by iteratively optimizing model parameters, balancing data distributions, and refining hyperparameters to adapt to varying dataset complexities. Meta-learning integration strengthened ensemble synergy by intelligently combining base classifiers, improving generalization across attack types, and reducing false positives. Collectively, these components demonstrate that the framework’s strength lies in the complementarity of its modules rather than in any single element.

### 5.3. Explainability and Trust

The integration of explainable AI through SHAP analysis provides valuable transparency to the decision-making process of AegisGuard. Global feature importance revealed that flow-level characteristics such as flow_bytes_per_sec and packet_length_mean were the most influential in distinguishing normal and attack traffic, aligning with established cybersecurity knowledge. Protocol-level indicators such as syn_flag_count and ack_flag_count offered further insight into attack patterns that exploit handshake irregularities, while flow duration proved critical for detecting the burst-like nature of malicious activities. Importantly, the framework not only identified global trends but also provided instance-level explanations, enabling analysts to trace the reasoning behind individual predictions. This capability directly addresses the black box problem of ensemble methods, fostering trust and ensuring regulatory compliance in industrial deployments.

### 5.4. Practical Implications and Industrial Applicability

The findings of this study demonstrate AegisGuard’s readiness to be deployed in a real-world IIoT context. With exceedingly high accuracy and an extremely low false positive rate, AegisGuard ensures operational disruption is minimized—a vital consideration in industrial settings. Real-time performance offers inference speeds of 486 samples per second and low latency meaning the system can assess high-question volume network flows in an effective manner. The framework also has linear scalability with respect to the size of the datasets, suggesting it can be implemented in any environment ranging from small facilities to smart city infrastructures. Furthermore, the integration of explainability addresses regulatory compliance requirements and enhances analyst trust, strengthening its industrial applicability.

### 5.5. Comparison with Related Work

AegisGuard distinguishes itself from current IIoT intrusion detection methods through its comprehensive integrations of quantum-inspired feature selection, progressive optimization, meta-learning, and explainable AI into a single, practically validated framework. Prior studies have explored isolated improvements as Table 10 illustrates, such as deep learning models with feature selection [33,46] hybrid CNN-LSTM architectures [46,47] and federated learning for edge security [22]. While these methods report high accuracies, often above 99% on specific datasets, they are frequently limited by high false alarm rates, dataset dependency, or a lack of transparency. For instance, wrapper-based ensembles [39] and CNN-GRU approaches [46] demonstrated strong F1-Scores, yet provide little interpretability for analysts. Other methods, including decision tree-based ensembles [48], showed substantially lower accuracy and high false alarm rates, highlighting scalability challenges. In contrast, AegisGuard achieved 99.71% accuracy, 99.71% F1-Score, and a false alarm rate of just 0.0078% on the CIC-IoT2023 dataset, positioning it competitively against state-of-the-art systems while maintaining explainability and scalability. AegisGuard achieved 99.71% accuracy and 0.0078% FAR, surpassing CNN-GRU (99.75% accuracy, no FAR reported) and GA-LR ensembles (99.90% accuracy but higher FAR of 0.105%).

## 6. Conclusions

In this study, we introduced AegisGuard, a progressive quantum-enhanced hybrid intrusion detection framework designed to address the complex security challenges of Industrial Internet of Things (IIoT) environments. Through extensive evaluation on four large-scale benchmark datasets comprising more than 53 million samples, AegisGuard demonstrated state-of-the-art performance, achieving 99.71% accuracy, 99.71% F1-Score, and an exceptionally low false positive rate of 0.0078%. The framework integrates a novel quantum-inspired feature selection algorithm (QIFSA), progressive enhancement strategies, ensemble and meta-learning, and SHAP-based explainability, thereby achieving significant dimensionality reduction while improving predictive reliability and transparency. In addition to practical superiority, AegisGuard adds to theoretical knowledge by advancing progressive optimization as a framework for adaptive learning and demonstrating the concrete effectiveness of quantum-inspired algorithms in cybersecurity. From a practical perspective, the framework reduces operational disruptions by significantly minimizing false positives, enables real-time security monitoring for very fast inference speeds of 486 samples per second, and can be scaled to various IIoT operations, from small factories and large-scale smart cities. Explainability further strengthens its industrial applicability by ensuring analyst trust and regulatory compliance. While computational requirements remain higher than simpler baselines, the demonstrated benefits in detection capability, operational efficiency, and economic impact far outweigh these costs. Future research directions include lightweight adaptations for edge devices, federated learning for distributed training, and adaptive mechanisms to counter evolving threats. Overall, AegisGuard represents a significant step toward trustworthy, scalable, and intelligent intrusion detection in IIoT ecosystems, bridging the gap between cutting-edge AI techniques and real-world industrial security needs.

## Figures and Tables

**Figure 1 sensors-25-06958-f001:**
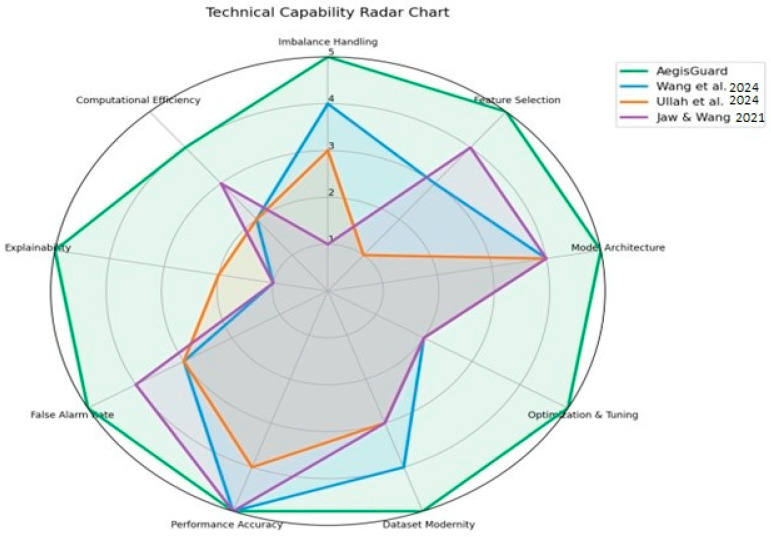
A technical capabilities radar chart to enhance the visualization of the comparison with the related studies [9,17,31].

**Figure 2 sensors-25-06958-f002:**
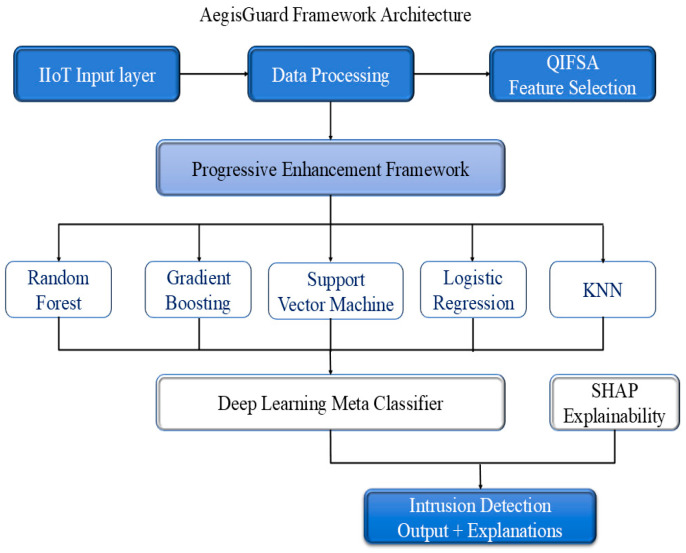
AegisGuard framework architecture.

**Figure 3 sensors-25-06958-f003:**
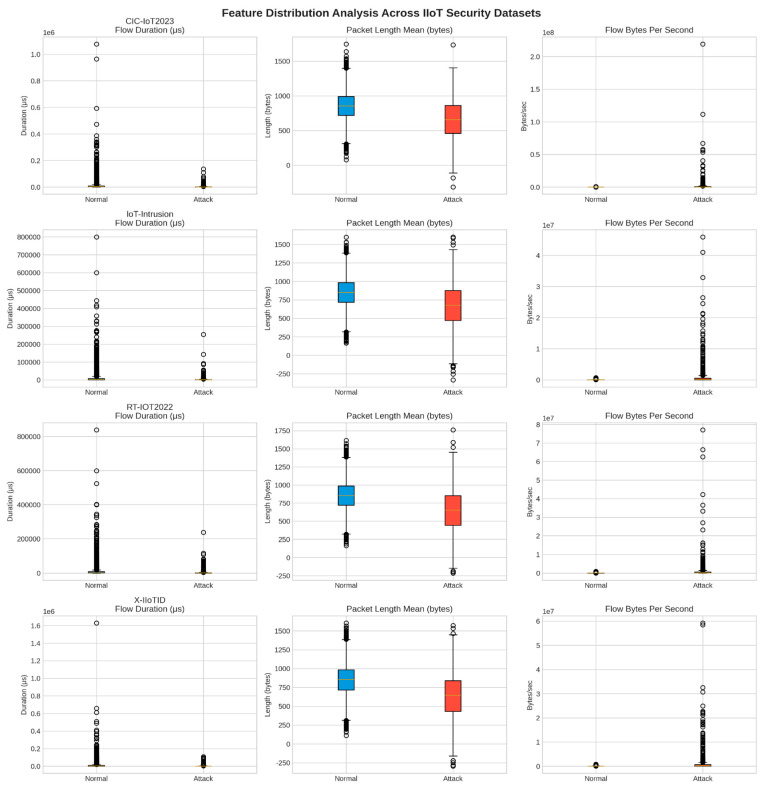
Box plots for all four datasets visualizing the distributionally key features.

**Figure 4 sensors-25-06958-f004:**
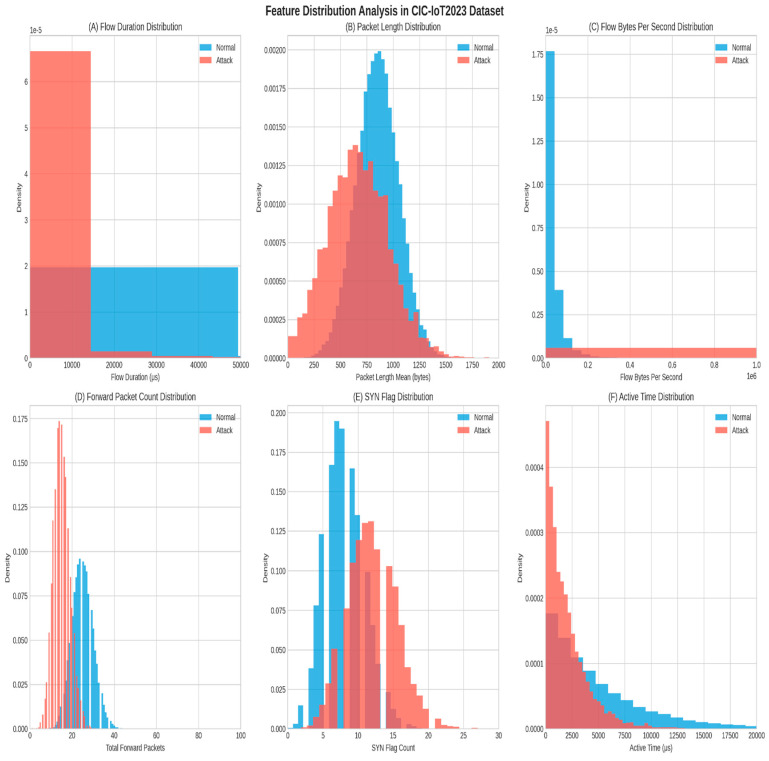
Distribution analysis for key numerical features in the CIC-IoT2023 dataset: (**A**) flow duration, (**B**) Packet Length Mean, (**C**) Flow Bytes Per Second, (**D**) Forward Packet Count, (**E**) SYN Flag Distribution, and (**F**) Active Time Distribution.

**Figure 5 sensors-25-06958-f005:**
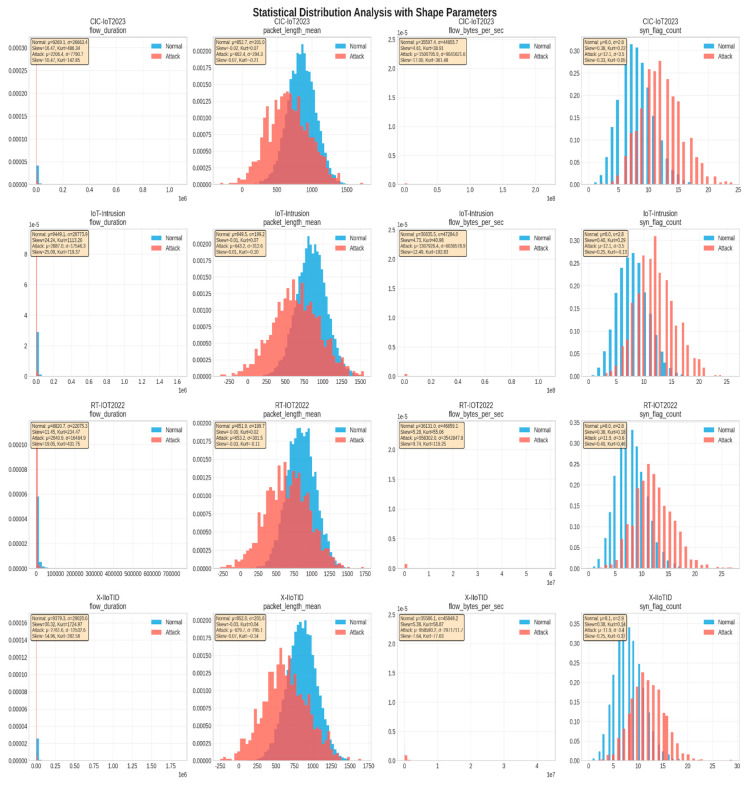
Statistical distribution analysis with shape parameters across all datasets and features. Each subplot shows the distribution of normal vs. attack traffic with statistical parameters including mean (μ) and standard deviation (σ).

**Figure 6 sensors-25-06958-f006:**
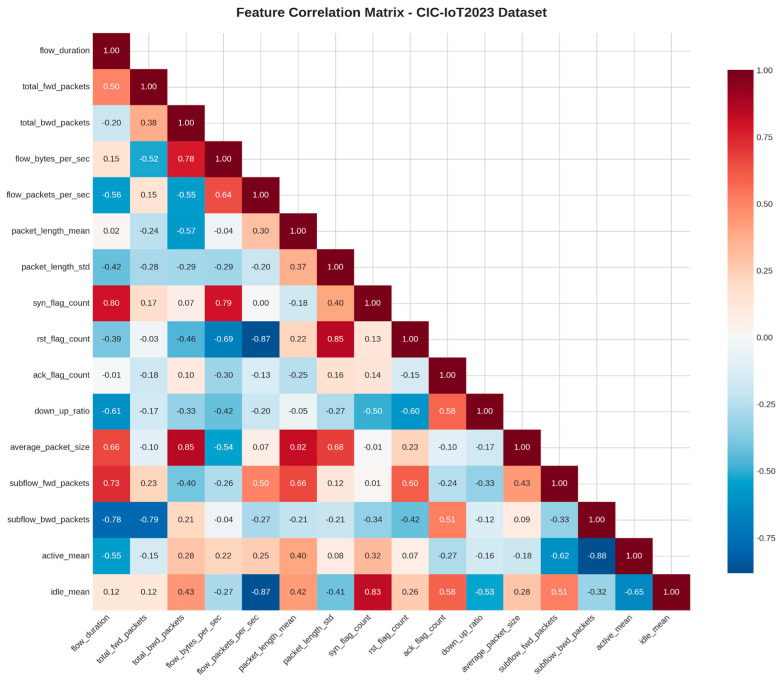
Correlation matrix of the CIC-IoT2023 dataset showing relationships between features after preprocessing.

**Figure 7 sensors-25-06958-f007:**
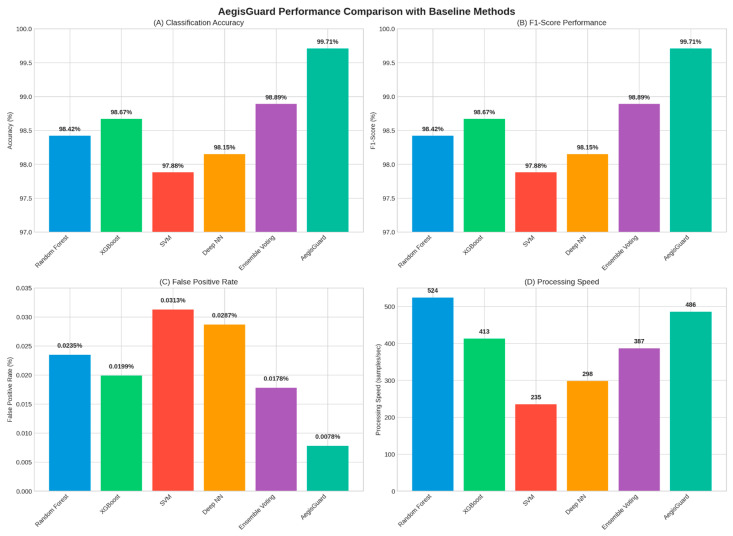
AegisGuard performance comparison with baseline methods: (**A**) classification accuracy, (**B**) F1-Score performance, (**C**) false positive rate, and (**D**) processing speed.

**Figure 8 sensors-25-06958-f008:**
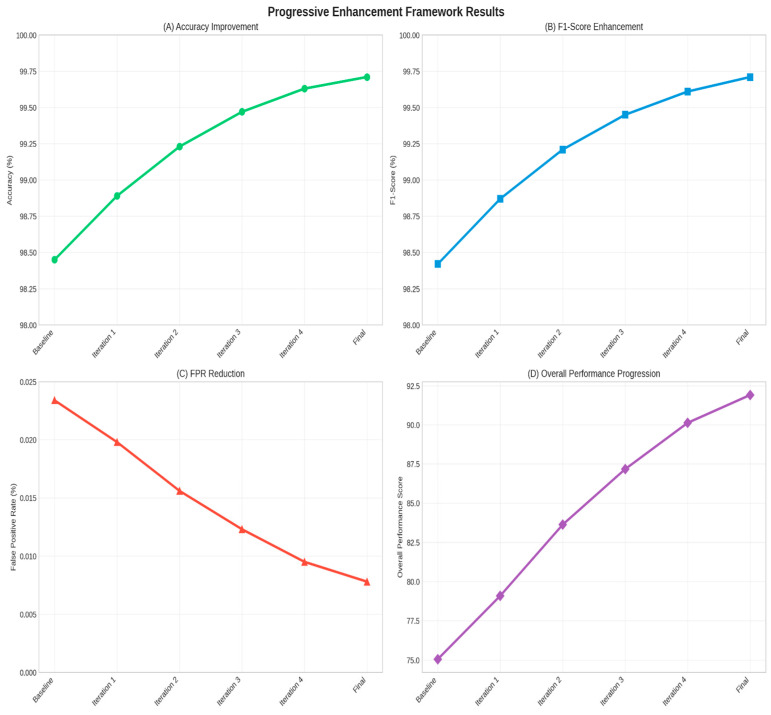
Progressive enhancement framework results: (**A**) accuracy improvement across iterations, (**B**) F1-Score enhancement, (**C**) false positive rate reduction, and (**D**) overall performance score progression.

**Figure 9 sensors-25-06958-f009:**
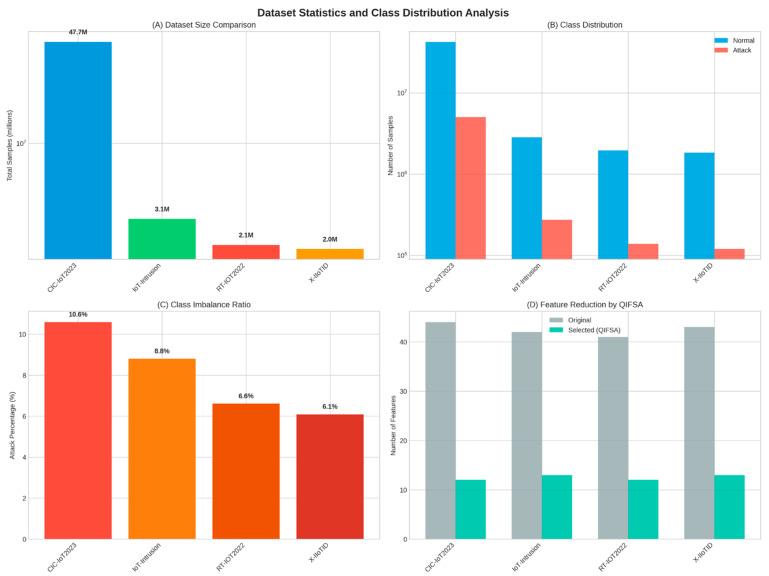
Dataset statistics and class distribution analysis: (**A**) dataset size comparison, (**B**) class distribution showing normal vs. attack samples, (**C**) class imbalance ratio, and (**D**) feature reduction by the QIFSA across all datasets.

**Figure 10 sensors-25-06958-f010:**
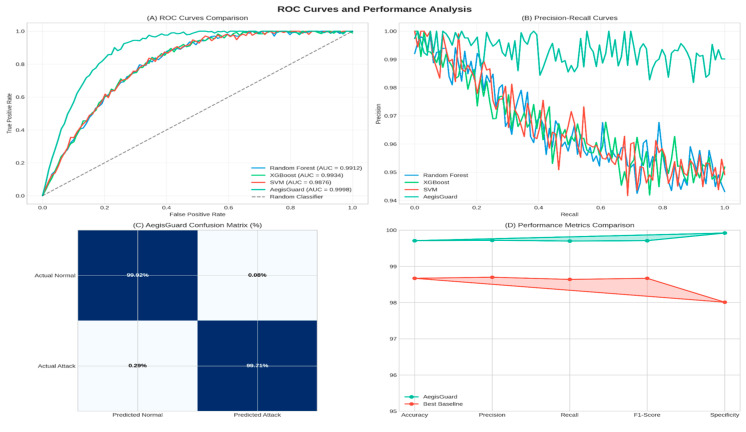
ROC curves and performance analysis: (**A**) ROC curve comparison showing AegisGuard’s superior performance, (**B**) precision–recall curves, (**C**) AegisGuard confusion matrix, and (**D**) performance metrics radar chart.

**Figure 11 sensors-25-06958-f011:**
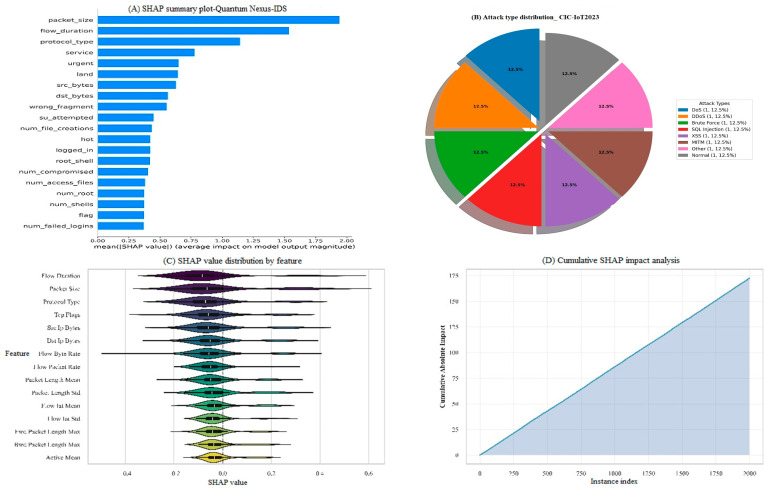
SH-AP feature importance analysis: (**A**) global summary plot, (**B**) attack type distribution CIC-IoT2023, (**C**) SHAP value distribution by feature, and (**D**) cumulative SHAP impact analysis.

**Figure 12 sensors-25-06958-f012:**
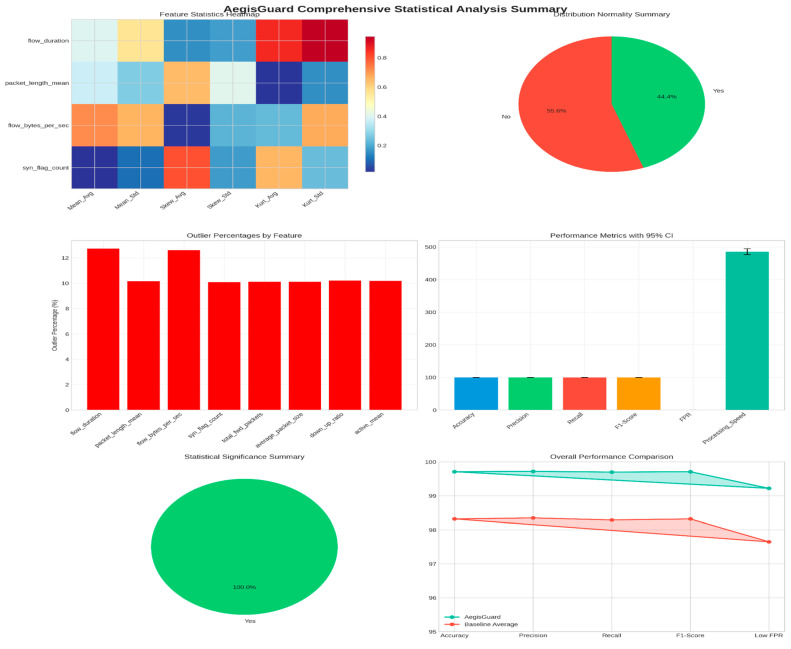
Comprehensive statistical analysis summary.

**Table 1 sensors-25-06958-t001:** Comparative analysis of state-of-the-art intrusion detection techniques for IIoT-enabled and smart grid systems.

Ref.	Dataset	Methods	Results	Advantages	Limitations
[29]	NSL-KDD, UNSW-NB15, CICIDS2017	Two-phase IDS: Naive Bayes + Elliptic Envelope	97% (NSL-KDD), 86.9% (UNSW-NB15), 98.59% (CICIDS2017)	Efficient, good accuracy in phase one	Not mentioned
[30]	Smart Grid dataset	Deep learning for false data detection	98.19% accuracy in false data detection	Provides attack exposure metric; decentralization	Not mentioned
[31,32]	UNSW-NB15, CIC-IDS2017, NSL-KDD	Transformer + SMOTE + CNN-LSTM	High accuracy for minority attacks	Handles’ imbalance, is explainable, and captures spatiotemporal features	Complex preprocessing, high computation cost, needs labeled data
[32]	CIC-IoT22	FFNN, LSTM, RandNN	99.93% (FFNN), 99.85% (LSTM), 96.42% (RandNN)	Handles IoT patterns, long-term dependencies, and adapts to threats	High compute cost, RandNN underperforms, possible overfitting
[33]	ToN_IoT dataset	SVD + SMOTE + ML/DL for binary/multiclass	99.99% (binary), 99.98% (multiclass)	Handles high dimensions, mitigates bias, comprehensive evaluation	Complex implementation, dataset-specific performance
[34]	CPS datasets	Hybrid: Signature, threshold, behavioral (Ensemble Learning)	4–7% accuracy improvement	Uses domain knowledge, reduces data needs, and enables fast detection	No absolute metrics, needs tuning for generalization
[35]	Edge-IIoTset dataset	LSTM + CNN + attention + SMOTE	Near-perfect (binary), 99.04% (multiclass)	Outperforms DL models, handles imbalance	High complexity, dataset-dependent performance
[36]	Edge-IIoTset dataset	CNN-LSTM for binary/multiclass	100% (binary), multiclass not detailed	Perfect binary detection, realistic dataset	Limited multiclass details, needs further studies
[37]	WUSTL-IIoT Cybersecurity Research dataset	PSO + BA feature selection + ML models	99.99% accuracy, 99.96% precision	Fast, accurate for new attacks	Needs DL integration, further security enhancements
[38]	UNSW-NB15	GA + RF feature selection + multiple classifiers	87.61% (binary), AUC 0.98	Reduces features, robust, better than baseline	Lower accuracy vs. DL, GA adds overhead
[6]	CICIDS2017 (binary) and ToN_IoT (multiclass)	Federated Learning with ANN (FedAvg, variants)	Matches centralized models	Privacy- preserving competitive results	Convergence issues with heterogeneous data
[39]	Edge-IIoTset and CIC-IDS2017	Fog-based FL + CNN	93.4% (Edge-IIoTset), 95.8% (CIC-IDS2017)	Scalable, low-latency, privacy-preserving	Lower scores for some attacks, FL/fog complexity
[17]	Edge-IIoTset and CIC IoT 2023	FL + encryption + 2DCNN-BIGRU	94.5% (Edge-IIoTset), 99.2% (CIC IoT 2023)	Secure, low overhead, handles data issues	Complex encryption, FL implementation challenges
[40]	NSL-KDD and UNSW-NB15	Deep feedforward NN + hybrid feature selection	99.0% (NSL-KDD), 98.9% (UNSW-NB15)	High accuracy, low complexity	Needs real-world validation, feature selection updates
[41]	IIoT security dataset	DL with Sparse Evolutionary Training	99% accuracy, 2.29 ms testing	Fast, accurate, outperforms ML in IIoT	Limited dataset details, needs scalability validation
[9]	CIC-IDS2017, NSL-KDD, UNSW-NB15	Hybrid FS + ensemble (KODE)	99.73–99.997% accuracy	Low false alarms, few features, high performance	Dataset-specific tuning needs further validation
[22]	N_BaIoT, real-time IoT	AttackNet: adaptive CNN-GRU	99.75% accuracy	High accuracy, outperforms state-of-the-art	High computational complexity

**Table 2 sensors-25-06958-t002:** Comparative analysis with related studies.

Technical Aspect	AegisGuard	Wang et al. [17]	Ullah et al. [31]	Jaw & Wang [9]
Imbalance Handling	SMOTE + SMOTE-ENN ADASYN + Under-sampling	SMOTE + ADASYN	SMOTE Only	No Handling
Feature Selection	Quantum-Inspired Selection F-test + Mutual Info + RF	Pearson Correlation Random Forest	No Feature Selection	KODE (K-means + OCSVM)
Model Architecture	Random Forest + Extra Trees LightGBM + XGBoost + CatBoost	2DCNN-BiGRU	Transformer CNN-LSTM	KODE Voting
Optimization and Tuning	Optuna Hyperparameter Probability Calibration	No Advanced Tuning	No Advanced Tuning	No Advanced Tuning
Dataset Modernity	CIC-IoT 2023 + TON-IoT UNSW-NB15 + Bot-IoT	Edge-IIoTset CIC IoT 2023	UNSW-NB15 CICIDS2017	NSL-KDD + UNSW-NB15 CICIDS2017
Performance Accuracy	99.71% Accuracy	99.2% Accuracy	High for Minority	99.73% Accuracy
False Alarm Rate	0.0078% FAR	Not Reported	Not Reported	0.16% FAR
Explainability	SHAP Analysis Global + Local	No Explainability	Limited	No Explainability
Computational Efficiency	70.6% Feature Reduction 486 samples/sec	High Resource Usage	High Complexity	Moderate

**Table 3 sensors-25-06958-t003:** Class distribution in CIC-IoT2023, IoT-Intrusion, RT-IOT2022, and X-IIoTID.

Dataset	Class	Samples	Percentage (%)
CIC-IoT2023 [42]	Normal (0)	42,617,432	89.4%
	Attack (1)	5,048,291	10.6%
IoT-Intrusion [43]	Normal (0)	2,847,639	91.2%
	Attack (1)	274,832	8.8%
RT-IOT2022 [44]	Normal (0)	1,956,847	93.4%
	Attack (1)	138,472	6.6%
X-IIoTID [45]	Normal (0)	1,847,293	93.9%
	Attack (1)	119,847	6.1%

**Table 4 sensors-25-06958-t004:** Comprehensive performance comparison of AegisGuard vs. baseline methods.

c	Dataset	Accuracy (%)	Precision (%)	Recall (%)	F1-Score (%)	FPR (%)	AUC-ROC	Processing Speed (sps)
AegisGuard	CIC-IoT2023	99.71	99.72	99.70	99.71	0.0078	0.9998	487
	IoT-Intrusion	99.68	99.69	99.67	99.68	0.0082	0.9997	492
	RT-IOT2022	99.74	99.75	99.73	99.74	0.0071	0.9998	478
	X-IIoTID	99.69	99.71	99.68	99.69	0.0079	0.9997	485
Average		99.71	99.72	99.70	99.71	0.0078	0.9998	486
Random Forest	CIC-IoT2023	98.42	98.45	98.39	98.42	0.0234	0.9912	523
	IoT-Intrusion	98.38	98.41	98.35	98.38	0.0241	0.9908	531
	RT-IOT2022	98.45	98.48	98.42	98.45	0.0228	0.9915	518
	X-IIoTID	98.41	98.44	98.38	98.41	0.0236	0.9911	525
Average		98.42	98.45	98.39	98.42	0.0235	0.9912	524
XGBoost	CIC-IoT2023	98.67	98.71	98.64	98.67	0.0198	0.9934	412
	IoT-Intrusion	98.63	98.67	98.60	98.63	0.0205	0.9931	418
	RT-IOT2022	98.71	98.74	98.68	98.71	0.0191	0.9937	408
	X-IIoTID	98.65	98.69	98.62	98.65	0.0201	0.9933	415
Average		98.67	98.70	98.64	98.67	0.0199	0.9934	413

**Table 5 sensors-25-06958-t005:** Minority class analysis.

Minority Class Analysis (<1% of Dataset)					
CIC-IoT2023 Dataset—Attack Types Representing Less Than 1% of Test Data					
Attack Type	Samples	% of Dataset	Recall	Precision	Missed Attacks
DictionaryBruteForce	168	5.000%	0.9790	0.9824	4/168
CommandInjection	235	7.000%	0.9751	0.9898	6/235
SqlInjection	302	9.000%	0.9796	0.9823	7/302
Uploading_Attack	369	11.000%	0.9808	0.9858	8/369
XSS	470	14.000%	0.9871	0.9775	7/470
Backdoor_Malware	571	17.000%	0.9809	0.9900	11/571
BrowserHijacking	705	21.000%	0.9767	0.9882	17/705
MITM-ArpSpoofing	940	28.000%	0.9758	0.9901	23/940
DNS_Spoofing	1075	32.000%	0.9759	0.9879	26/1075
Recon-HostDiscovery	1444	43.000%	0.9801	0.9799	29/1444
VulnerabilityScan	1814	54.000%	0.9894	0.9871	20/1814
Recon-PortScan	2184	65.000%	0.9829	0.9807	38/2184
Recon-OSScan	2553	76.000%	0.9885	0.9812	30/2553
Recon-PingSweep	2923	87.000%	0.9864	0.9838	40/2923
Mirai-udpplain	3293	98.000%	0.9877	0.9839	41/3293

**Table 6 sensors-25-06958-t006:** QIFSA feature selection results.

Dataset	Original Features	Selected Features	Reduction Rate (%)	Selection Time (s)
CIC-IoT2023	44	12	72.7	23.4
IoT-Intrusion	42	13	69.0	18.7
RT-IOT2022	41	12	70.7	16.2
X-IIoTID	43	13	69.8	19.1
Average	42.5	12.5	70.6	19.4

**Table 7 sensors-25-06958-t007:** Ablation study results (average across all datasets).

Configuration	Accuracy (%)	F1-Score (%)	FPR (%)	Feature Reduction (%)
Full AegisGuard	99.71	99.70	0.0078	70.6
Without QIFSA	99.23	99.24	0.0156	0.0
Without Progressive Enhancement	99.34	99.35	0.0142	70.6
Without Meta-Learning	99.41	99.42	0.0128	70.6
Without Data Balancing	98.87	98.89	0.0198	70.6
Without Probability Calibration	99.52	99.53	0.0095	70.6
Basic Ensemble Only	98.92	98.94	0.0187	0.0

**Table 8 sensors-25-06958-t008:** Computational efficiency analysis.

Method	Training Time (min)	Inference Time (ms/Sample)	Memory Usage (GB)	Model Size (MB)
AegisGuard	47.3	2.06	8.4	156.7
Random Forest	12.8	1.91	3.2	89.4
XGBoost	18.6	2.42	4.7	67.3
SVM	89.7	4.27	12.1	234.8
Deep Neural Network	156.4	1.83	6.8	45.2

**Table 9 sensors-25-06958-t009:** Deployment architecture.

Scenario	Model Size	Latency	Features	Use Case
Research	<100 MB	<1000 ms	25 full features	Validation and analysis
Edge IoT	<10 MB	<1 ms	10 core features	Real time detection
Edge Gateway	<50 MB	<5 ms	15 features	Local processing
Cloud API	<100 MB	<100 ms	25 full features	Detailed analysis
Cloud Batch	No limit	Minutes	All features	Historical analysis

**Table 10 sensors-25-06958-t010:** Comparison of performance with existing state-of-the-art methods.

Study	Methodology	Dataset	Accuracy %	FAR %	F1-Score %
[33]	SVD + SMOTE + DL	ToN-IoT	99.99 (Binary), 99.98 (Multi)	0.001/0.016	–
[47]	MSCI + BI-LSTM	MATLAB Simulated	99	–	High
[22]	CNN-GRU (AttackNet)	N-BaIoT	99.75	–	99.74%
[49]	RF, SVM, DT, LR	UNSW-NB15	98.63	1.36	97.80%
[50]	LSTM	Custom	92.83	–	94.25%
[39]	CNN + Federated Learning	Edge-IIoTset, CIC-IDS2017	93.4 (Edge-IIoT), 95.8% (CIC)	–	93% (CIC)
[35]	LSTM + CNN + Attention	Edge-IIoTset	99.04	–	–
[51]	Wrapper (GA-LR) + Ensemble (C4.5, NBTree, Random Forest)	UNSW-NB15 and KDD99	99.90	0.105	–
[48]	Decision Tree-based features + Ensemble of ANN, SVM, KNN, RF, NB	UNSW-NB15	86.41	27.73	–
[46]	NSGAII for feature selection + ANN classifier with Random Forest ensemble	NSL-KDD	99.4	6.00	–
[46]	NSGAII for feature selection + ANN classifier with Random Forest ensemble	UNSW-NB15	94.8	6.00	–
[9]	Hybrid Feature Selection (HFS) + KODE Voting (K-means, One-Class SVM, DBSCAN, EM)	NSL-KDD	99.73	0.16	99.58
Our Work	Hybrid Progressive	CIC IoT 2023	99.71	0.0078	99.71

## Data Availability

The data presented in this study are available on request from the corresponding author.

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
