# Peer review of "AegisGuard: A Multi-Stage Hybrid Intrusion Detection System with Optimized Feature Selection for Industrial IoT Security"

_sensors, 2025, doi:10.3390/s25226958_

Round 1
Reviewer 1 Report
Comments and Suggestions for Authors
In my assessment, the paper tackles a very important problem but unfortunately falls short in several key areas that are critical for publication. From my perspective, the core issue is that I finished reading without a clear sense of what new knowledge this work provides.
1. My View on Novelty and Contribution:
Personally, I struggled to identify the novel core of this paper. It feels like a composition of existing, well-known techniques—SMOTE variants, standard feature selection, and powerful but standard ensemble models. In my opinion, the manuscript would be significantly strengthened by a much clearer and earlier statement that distinguishes its unique contribution from simply applying these tools competently to a new dataset.
2. My Reaction to the Abstract:
I found the abstract quite difficult to parse. The high density of acronyms and methodological jargon made it hard for me to quickly grasp the research's objective and significance. I believe an abstract should prioritize clarity and narrative flow, saving the deep technical specifics for the main body of the paper. This would make the work more accessible to a broader audience.
3. My Thoughts on the Introduction:
From my point of view, the introduction begins too technically. I would have preferred it to first walk me through the general landscape of IoT security challenges and the persistent issue of class imbalance, before introducing the specific technical approach. The early listing of algorithms felt premature and made it harder for me to understand the motivation behind the work.
4. My Perspective on the Literature Review:
I felt the literature review was more descriptive than analytical. It tells me what others have done, but often doesn't provide a clear opinion on what they failed to do or where the weaknesses in prior approaches lie. I was left wanting a more guided critique of the existing literature that would logically lead to and justify the authors' own approach.
5. My Concerns on the Methodology:
My main concern here is a lack of detail that would allow for reproducibility. I am unsure about the exact data splitting strategies and preprocessing steps for each distinct dataset. Furthermore, I was very surprised to see an unmodified, AI-generated figure. In my opinion, this significantly undermines the section's credibility, as it suggests a lack of original thought in presenting the methodological framework.
6. My Analysis of the Experimental Results:
I remain unconvinced by the results section. While the accuracy is high, I am of the opinion that for imbalanced problems like this, metrics like per-class precision, recall, and F1-score are non-negotiable and tell the real story. Without them, it's difficult to trust that the model isn't just ignoring the minority classes. Most importantly, I really wanted to see an ablation study. Without it, we have to take it on faith that the entire complex pipeline is necessary, when in my view, the strong results could potentially be driven by a single component, like the use of a powerful classifier like LightGBM.
The English language quality in the provided peer review feedback is of a high standard, demonstrating clarity, precision, and a formal academic tone suitable for scholarly communication. The critique employs a sophisticated vocabulary, including terms like "articulate," "methodological jargon," and "ablation study," which are appropriate for the context. The sentences are well-structured and grammatically sound, flowing logically to build a coherent argument. While the original text contained informal phrasing and grammatical errors, the refined response effectively maintains a professional and objective tone, focusing on constructive criticism. It successfully adheres to the conventions of academic writing by being both critical and respectful, ensuring the feedback is clear and actionable for the authors.
Author Response
Please find enclosed the updated

Reviewer 2 Report
Comments and Suggestions for Authors
The article is written on the current topic of protecting the infrastructure of intelligent networks. It is presented in a scientific style, is well structured and contains a sufficient number of experimental results. The authors proposed an innovative solution in the form of a design that includes a four-stage hybrid sampling container designed to select parameters and adjust them to ensure the accuracy of detecting unauthorized intruders. The novelty of the model lies in the fact that it provides protection of the critical infrastructure of IIoT facilities based on modern optimization methods. There are some technical errors. After correcting which, the article is recommended for publication.
Remarks.
- There are technical design errors. For example, according to the link to Figure 1 (line 190), the AegisGuard Framework, the architecture of which is designed to solve problems of unauthorized access in IIoT, should be described in more detail. Also, the name of the figure is missing in line 217-218.
- It is advisable to explain in more detail the advantages of the results obtained on the basis of the correlation matrix (Figure 4, line 433-434) and the practical value of visualization.
- In Figure 9, the format of the fragment representation under the letter "C" should be considered. Line 544. Because compared to other fragments, this part loses its appeal.
Author Response
Research Article:
AegisGuard: A progressive Quantum-Enhanced Hybrid Intrusion Detection System for Industrial Internet of Things Security
Manuscript ID: Sensors-3900177
|
Response to Reviewer 2 Comments
|
||
|
1. Summary |
|
|
|
We would like to express our sincere gratitude for the time and effort you dedicated to reviewing our manuscript. Below, we provide detailed responses to each of your comments, along with the corresponding revisions, which have been highlighted in the re-submitted manuscript files.
|
||
|
2. Questions for General Evaluation |
Reviewer’s Evaluation |
Response and Revisions |
|
Does the introduction provide sufficient background and include all relevant references?
|
|
Yes |
|
Are all the cited references relevant to the research?
|
|
Yes |
|
Is the research design appropriate?
|
|
Yes |
|
Are the methods adequately described?
|
|
Can be improved
|
|
Are the results clearly presented? |
|
Can be improved
|
|
Are the conclusions supported by the results? |
|
Yes
|
|
3. Point-by-point response to Comments and Suggestions for Authors |
||
|
Comment 1: [ There are technical design errors. For example, according to the link to Figure 1 (line 190), the AegisGuard Framework, the architecture of which is designed to solve problems of unauthorized access in IIoT, should be described in more detail. Also, the name of the figure is missing in line 217-218.] |
||
|
Response 1: Thank you for pointing this out. We agree with this comment, and we have added the label of the mentioned figure (section 3, page 9, line 278). Furthermore, we extended the description of the AegisGuard Framework to give an overview about the proposed architecture(section 3, page 8, line 262-273). |
||
|
Comment 2: [ It is advisable to explain in more detail the advantages of the results obtained on the basis of the correlation matrix (Figure 4, line 433-434) and the practical value of visualization] |
||
|
Response 2: Agree. We have, accordingly, modified the correlation analysis section (section 3.5, pages 18-19, Lines 545-559) to further detail the correlation matrix analysis for the results obtained and to show the visualization value.
Comment 3: [In Figure 9, the format of the fragment representation under the letter "C" should be considered. Line 544. Because compared to other fragments, this part loses its appeal.] Response 3: Thank you for raising this comment. Regarding the format of the mentioned (figure 10, page 28), we enhanced the figure format as much as possible.
|
||
Please find enclosed the updated research document incorporating the requested changes. Thank you for your valuable input and collaboration

Reviewer 3 Report
Comments and Suggestions for Authors
This paper presents an advanced hybrid intrusion detection system, AegisGuard, designed specifically for IIoT environments such as smart grids. The integration of a four-stage sampling pipeline combining methods like SMOTE, SMOTEEN, ADASYN, and strategic undersampling is a noteworthy approach to address the severe class imbalance prevalent in IIoT traffic datasets. Additionally, the paper’s innovative use of a quantum-inspired feature selection scheme, which fuses statistical significance, information content, and ensemble importance with a trust-aware weighting mechanism, adds a novel dimension to feature reduction methods. The comprehensive evaluation across multiple datasets, demonstrating improvements in accuracy and false positive reduction, is a significant strength of the work.
However, despite these promising aspects, the manuscript currently lacks sufficient technical detail and clarity. The description of the quantum-inspired feature selection process, particularly how the multidimensional scores are computed, normalized, and combined, is somewhat superficial. A deeper explanation, with clear mathematical formulations and a step-by-step illustration, would clarify the methodology and enable reproducibility. Furthermore, the experimental setup and results section need more comprehensive analysis, including ablation studies that isolate the impact of each component of the proposed framework, such as the sampling pipeline, feature selection, and classifier ensemble. Without this, it is difficult to fully assess the contributions of each element to the overall performance.
The references need to be accurate, up-to-date, and comprehensive, ensuring all related work is appropriately acknowledged and contextualized. For instance, it is necessary to discuss side-channel attacks and its defenses in Section 1 and Section 2. Below are several papers about side-channel attacks and corresponding authentication methods you are suggested to add into the next version.
- Side-channel attacks in IoT: (1) Uncovering User Interactions on Smartphones via Contactless Wireless Charging Side Channels (S&P’23), (2) FOAP: fine-grained open-world android app fingerprinting (USENIX Security’22), (3) Packet-level open-world app fingerprinting on wireless traffic (NDSS’22)
- Authentication methods in IoT: (1) HandKey: Knocking-triggered robust vibration signature for keyless unlocking (TMC’22), (2) MagSign: Harnessing Dynamic Magnetism for User Authentication on IoT Devices (TMC’23), (3) LiveProbe: Exploring Continuous Voice Liveness Detection via Phonemic Energy Response Patterns (IoTJ’22)
In addition, the writing could benefit from improved organization and coherence. The manuscript jumps between various aspects—dataset descriptions, technical methods, and system architecture—without sufficiently guiding the reader through the narrative. A more structured presentation, with dedicated subsections for the methodology, experimental setup, and results discussion, would enhance readability and clarity. Lastly, there is limited discussion of the practical deployment considerations, such as computational requirements, real-time scalability, and robustness in dynamic environments. Addressing these aspects would not only strengthen the paper but also demonstrate its practical relevance for real-world IIoT security.
Author Response
|
1. Summary |
|
|
|
Thank you very much for taking the time to review this manuscript. Please find the detailed responses below and the corresponding revisions/corrections highlighted/in track changes in the re-submitted files. |
||
|
2. Questions for General Evaluation |
Response and Revisions |
|
|
Does the introduction provide sufficient background and include all relevant references?
|
Yes/Can be improved/Must be improved/Not applicable |
|
|
Are all the cited references relevant to the research?
|
Yes/Can be improved/Must be improved/Not applicable |
|
|
Is the research design appropriate?
|
Yes/Can be improved/Must be improved/Not applicable |
|
|
Are the methods adequately described?
|
Yes/Can be improved/Must be improved/Not applicable |
|
|
Are the results clearly presented? |
Yes/Can be improved/Must be improved/Not applicable |
|
|
Are the conclusions supported by the results?
|
Yes/Can be improved/Must be improved/Not applicable |
|
|
3. Point-by-point response to Comments and Suggestions for Authors
|
||
|
Comments 1: [ The description of the quantum-inspired feature selection process, particularly how the multidimensional scores are computed, normalized, and combined, is somewhat superficial. A deeper explanation, with clear mathematical formulations and a step-by-step illustration, would clarify the methodology and enable reproducibility. ]
|
||
|
Response 1: Agree, and we added a dedicated discussion section (section 3.3.1, page 15-16, lines 470-495) for QIFSA, including the mathematical formulas and tying them back to the operational logic of QIFSA.
Comments 2: [ the experimental setup and results section needs more comprehensive analysis, including ablation studies that isolate the impact of each component of the proposed framework, such as the sampling pipeline, feature selection, and classifier ensemble. ]
Response 2: Thank you for pointing this out. An ablation study results is presented in Table 5, included in the ROC Analysis and Performance Metrics section (section 4.6, pages 25-27, lines 663-692 ) showing the impact of isolating each component of the proposed framework.
|
||
|
Comments 3: [ The references need to be accurate, up-to-date, and comprehensive, ensuring all related work is appropriately acknowledged and contextualized. For instance, it is necessary to discuss side-channel attacks and its defenses in Section 1 and Section 2. Below are several papers about side-channel attacks and corresponding authentication methods you are suggested to add into the next version. - Side-channel attacks in IoT: (1) Uncovering User Interactions on Smartphones via Contactless Wireless Charging Side Channels (S&P’23), (2) FOAP: fine-grained open-world android app fingerprinting (USENIX Security’22), (3) Packet-level open-world app fingerprinting on wireless traffic (NDSS’22)
- Authentication methods in IoT: (1) HandKey: Knocking-triggered robust vibration signature for keyless unlocking (TMC’22), (2) MagSign: Harnessing Dynamic Magnetism for User Authentication on IoT Devices (TMC’23), (3) LiveProbe: Exploring Continuous Voice Liveness Detection via Phonemic Energy Response Patterns (IoTJ’22) ]
|
||
|
Response 3: Agree. We really value your comment, and we have extended the discussion of the introduction section (section 1, page 3 , lines 75-84 ) and the related works section (section 2, page 5 , lines 212-229 ) to cover side-channel attacks and IoT devices Authentication based on the recommended references.
Comments 4: [ The manuscript jumps between various aspects—dataset descriptions, technical methods, and system architecture—without sufficiently guiding the reader through the narrative. A more structured presentation, with dedicated subsections for the methodology, experimental setup, and results discussion, would enhance readability and clarity. ]
Response 4: Thank you for providing this comment. Regarding the structure of the sections, we improved the paper structure to improve clarity.
Comments 4: [ Lastly, there is limited discussion of the practical deployment considerations, such as computational requirements, real-time scalability, and robustness in dynamic environments. Addressing these aspects would not only strengthen the paper but also demonstrate its practical relevance for real-world IIoT security. ] Response 4: We really appreciate your concern about practical deployment considerations. Hence, we added a section (section 4.8.3, page 21) including a table to show the multi-scenario deployment architecture. |
||
|
4. Response to Comments on the Quality of English Language: Done |
||
|
5. Additional clarifications: Nothing more thanks |
||
|
|
||
Please find enclosed the updated research document incorporating the requested changes. Thank you for your valuable input and collaboration

Round 2
Reviewer 1 Report
Comments and Suggestions for Authors
The authors have put a lot of effort into dealing with all reviewer comments. The revised paper is much clearer, better structured, and more analytical than before. The Introduction is now a comprehensive overview of the security challenges surrounding IoT and IIoT before the proposed framework is introduced. The Related Work section has undergone a complete overhaul, making it more analytical, and explicitly pointing out the shortcomings of previous studies as well as the areas in which the proposed AegisGuard framework does better. The Methodology section is now longer, with clearer descriptions of data preprocessing, sampling, and feature selection steps that increase reproducibility. The authors have not only discarded the AI-generated figure but also presented an original and artistic framework diagram, thereby bringing about an improvement in both credibility and presentation quality. In the Results and Discussion sections, the reader can now find detailed performance metrics per class as well as a comprehensive ablation study that together showcase the robustness and balanced performance of the proposed model even amidst the majority and minority attack classes. In all, the paper is a result of extensive revisions and high technical rigor. The authors are commended for taking the feedback seriously and for making the paper more readable, reproducible, and scientifically contributive. My recommendation is that the paper is ready for publication after just minor editorial polishing.
Author Response
Comment 1 : [ My recommendation is that the paper is ready for publication after just minor editorial polishing.]
Response: We are grateful to the reviewer for the positive evaluation and encouraging feedback. In accordance with this recommendation, we carefully reviewed the entire manuscript and performed minor editorial polishing to improve clarity, grammar, and consistency.

Reviewer 3 Report
Comments and Suggestions for Authors
I appreciate the authors for the submission and the revisions. Most of my previous concerns were addressed with some other suggestions.
- It still lacks references about IoT authentications and networks, such as the following studies.
Chen, Y., Ni, T., Xu, W. and Gu, T., 2022. SwipePass: Acoustic-based second-factor user authentication for smartphones. Proceedings of the ACM on Interactive, Mobile, Wearable and Ubiquitous Technologies, 6(3), pp.1-25.
Ni, T., Lan, G., Wang, J., Zhao, Q. and Xu, W., 2023. Eavesdropping mobile app activity via {Radio-Frequency} energy harvesting. In 32nd USENIX Security Symposium (USENIX Security 23) (pp. 3511-3528).
Xu, W., Lan, G., Lin, Q., Khalifa, S., Bergmann, N., Hassan, M. and Hu, W., 2017. Keh-gait: Towards a mobile healthcare user authentication system by kinetic energy harvesting. In Proceedings of the 24th Annual Network and Distributed System Security Symposium, NDSS 2017. The Internet Society.
